# Intention, Motivation, and Empowerment: Factors Associated with Seasonal Influenza Vaccination among Healthcare Workers (HCWs)

**DOI:** 10.3390/vaccines11091508

**Published:** 2023-09-21

**Authors:** Thalia Bellali, Polyxeni Liamopoulou, Savvato Karavasileiadou, Noura Almadani, Petros Galanis, George Kritsotakis, Georgios Manomenidis

**Affiliations:** 1Department of Health Sciences, School of Sciences, European University Cyprus, Nicosia 2404, Cyprus; 2Faculty of Nursing, International Hellenic University, 57400 Thessaloniki, Greece; tzeniliam@gmail.com; 3Department of Community Health Nursing, College of Nursing, Princess Nourah bint Abdulrahman University, Riyadh 11671, Saudi Arabia; skaravasileiadou@pnu.edu.sa (S.K.); naalmadani@pnu.edu.sa (N.A.); 4Faculty of Nursing, National and Kapodistrian University of Athens, 11524 Athens, Greece; pegalan@nurs.uoa.gr; 5Department of Business Administration & Tourism, Hellenic Mediterranean University, 72300 Herakleion, Greece; gkrits@hmu.gr; 6Faculty of Nursing, International Hellenic University, Didimoteicho Branch, 57400 Thessaloniki, Greece; george.mano@yahoo.gr

**Keywords:** behavior, healthcare workers, influenza, vaccination, factors

## Abstract

Background: Vaccination against seasonal influenza has proven effective in preventing nosocomial influenza outbreaks among hospital patients and healthcare workers (HCWs). This study aims to explore the intention, motivation, and empowerment toward vaccination and vaccination advocacy as contributing factors for seasonal influenza vaccination in HCWs. Methods: A cross-sectional study in eight secondary hospitals in Greece was conducted from March to May 2022. An anonymous questionnaire was enclosed in an envelope and distributed to all participants, including questions on vaccine behavior and the MoVac-flu and MoVad scales. Results: A total of 296 participants completed the questionnaire. In multivariate logistic regression models adjusted for potential confounders, increased age, intention score, MoVac-flu scale score, and the presence of chronic diseases were significant predictors of influenza vaccination this year, while increased age, intention score, and presence of chronic diseases were predictors of vaccination every year. Conclusion: Vaccination uptake is simultaneously affected by logical cognitive processes (intention), together with factors related to motivation and empowerment in distinct self-regulatory domains such as value, impact, knowledge, and autonomy. Interventions focused on these identified predictors may be used as a guide to increase HCWs’ vaccination rates.

## 1. Introduction

Healthcare workers (HCWs) are at a high risk of becoming infected with seasonal influenza infection and subsequently infecting their patients due to direct patient care and the asymptomatic character of the influenza virus. Potentially severe complications of influenza infection may lead to increased morbidity and mortality in patients with additional underlying medical conditions [1]. According to the World Health Organization (WHO), up to 500,000 people die because of influenza complications [2], while the annual economic burden is estimated to be around USD 11.2 billion in the USA [3] and EUR 77 million on hospitalizations in Europe [4]. Economic hardships caused by work absences and low productivity are an additional health system burden [5]. Vaccination is considered the most effective tool for managing the spread of this infection [6,7].

Despite the WHO’s recommendations for annual vaccination [2], the rates of vaccine uptake among European HCWs, according to the European Center for Disease Prevention and Control (ECDC), vary distinctively [8] and are largely poor, with a coverage rate of 25.7% (5.0% to 54.4%) for the 2014-15 season [9], if one considers the aim of 75% uptake for people aged 65 recommended by the WHO [2]. In France, nurses had the least favorable opinions about seasonal influenza vaccination (SIV) compared to other vaccines. Only one-third were vaccinated in one season [10,11]. In Greece, SIV uptake is estimated at approximately 24.7% [12], although higher percentages (57%) have been reported for general practitioners [13]. In line with these results, Kopsidas et al. [14] said that only 24% of nurses and 60% of doctors were vaccinated in a tertiary pediatric hospital.

The introduction of a previously unknown new virus named SARS-CoV-2 at the beginning of 2020 played a significant part in reducing the circulation of the influenza virus, disrupting the spread of the disease mainly because of the use of face masks. The focus of public attention on COVID-19 led to a lower demand for the influenza vaccine and a decline in flu vaccination due to the convergence of influenza and COVID-19 vaccinations, as according to guidelines, a space of at least 2 weeks is suggested. However, in certain age groups (i.e., elderly), vaccination uptake remained stable [15].

Many factors have been shown to determine influenza vaccination uptake. In an Italian study of HCWs in various healthcare facilities, chronic illnesses, and smoking habits were pivotal factors for SIV [16]. Several studies have also associated the following demographic variables with HCWs’ intention to receive the seasonal influenza vaccination: age, gender, and profession. Male and older HWCs were more likely to receive the SIV [10,13,17], and nurses displayed lower intention to receive the SIV when compared to physicians [18].

As mentioned above, the main shortcoming of those studies is that they need to rely on a theory to interpret their findings. Among the early but still relevant theories trying to explain the underlying reasons for human behavior in a particular context and a specific time (in this case, vaccination uptake) are the ‘Theory of Reasoned Action’(TRA) and the ‘Theory of Planned Behavior’ (TPB) [19,20].

The ‘TPB’, proposed by Ajzen [20,21], is based mainly on Ajzen and Fishbein’s ‘TRA’ [22], where an individual who intends to perform a behavior has more chances to act accordingly. People do not make health-related decisions in a vacuum but rather in a particular physical, socio-cultural, and political context with specific barriers and facilitators [23,24,25,26]. To overcome this barrier, a perceived behavior control was added as a significant variable to the ‘TRA’, forming the ‘TPB’ [21].

Based on the above theories, extensive efforts have been made to define the determinants of HCWs’ intention to receive the seasonal influenza vaccination. A study conducted in Singapore on HCWs working in an acute care tertiary hospital investigated the association between psychosocial beliefs and vaccine uptake using the Health Belief Model-‘HBM’, an earlier theory of the ‘TRA’ [25,27]. It concluded that the higher HCWs’ perceptions of vaccine effectiveness, the higher the intention for vaccine uptake [28]. In Europe, Boey et al. [29] surveyed HCWs from various hospitals and nursing homes in Belgium using the Health Belief and additional models [22,30]. They found that HCWs’ efforts to protect their families and environments were essential factors that affected vaccination behavior. All these studies were based on theories that tried to explain HCWs’ beliefs on influenza vaccine uptake based on their rational consideration of the cost and benefits of this action [31].

People do not always make rational and coherent choices but rather choices based on the norms of the broader social environment and their emotions [23,24,32]. As such, they may engage (or not) in a behavior because other people do it because of how they feel or as a coping mechanism, not because it is the right thing to do or the wrong thing to avoid [33]. The Cognitive Model of Empowerment (CME) offers a basis for investigating vaccine and health-related behaviors [34,35,36]. The ‘CME’ states that individuals engage in conduct not only based on their rational thoughts but also based on their feelings in four distinct domains: value; impact; autonomy and knowledge [36].

A recent study tried to advance our understanding of the determinants of HCWs’ intention to receive the SIV using the ‘CME’ [35]. The authors examined the feelings that determine why HCWs may want to get vaccinated in six European countries, in addition to explaining influenza vaccination uptake as a weighted process of a cost–benefit action [34,35,36]. They concluded that self-perceived efficacy was the primary factor for vaccination advocacy from the HCWs.

The present study aimed to simultaneously test the predictive value of the ‘TPB’ and the ‘CME’ to explore intention, motivation, and empowerment towards influenza vaccination and vaccination advocacy as contributing factors to seasonal influenza vaccination. In this way, we can examine the possible intercorrelation of rational reasoning and emotional factors in vaccine uptake.

## 2. Materials and Methods

### 2.1. Study Design and Population

A cross-sectional study was conducted between March and May 2022. The target population was a convenience sample of HCWs in 8 secondary hospitals in a northern province of Greece (response rate 74%). Inclusion criteria were having at least one year of experience and permanent employment status. Posters describing the study were placed outside the nursing stations of all hospital departments. An anonymous questionnaire explaining the study’s purpose and contact information was enclosed in an envelope and distributed to all participants. They were then asked to return it sealed in a general mailbox placed in every department. The time for completion was approximately 15–20 min.

### 2.2. Measures

Participants reported their influenza vaccination status this year and if they received the seasonal influenza vaccine every year. They also provided information on the below topics.

#### 2.2.1. Demographics

Demographic data included participants’ gender, age, marital status, number of children, profession (physicians, nurses, nurse assistants), working department, level of education, and years of working experience.

#### 2.2.2. Intention for Vaccination Uptake Based on the Theory of Planned Behavior

A questionnaire constructed based on the ‘TPB’ guidelines assessed the intention for vaccination uptake. It was previously used in a Greek elderly population [37] and was then adapted for HCWs. The questionnaire consists of four subscales that correspond to attitudes (3 items), subjective norms (6 items), perceived behavioral control (4 items), and intention (2 items). The attitudes subscale defines attitudes towards vaccination uptake. The subjective norms subscale includes statements concerning the expectation of HCWs’ social context to uptake vaccination. The perceived control subscale reflects individual control in receiving the vaccine. Finally, the intention scale assesses the desire for vaccination. All items were measured on a 7-point Likert scale ranging from 1, “strongly disagree,” to 7, “strongly agree”. A higher score indicates positive attitudes and beliefs regarding influenza vaccination, higher social pressure, higher possibility of getting a vaccine shot, etc. To ensure the quality of the adapted version for HCWs, the questionnaire was distributed through a pilot study to a group of twenty nurses/nurse assistants and ten physicians who were asked about inconsistencies. The face validity of the questionnaire was excellent since there were no comments/questions/misunderstandings. The Cronbach’s alpha coefficients for the four subscales ranged from a = 0.73 (perceived control scale) to a = 0.95 (intention scale), indicating acceptable to excellent reliability.

#### 2.2.3. Cognitive Empowerment towards Influenza Vaccination and Vaccination Advocacy

Cognitive empowerment towards influenza vaccination and vaccination advocacy were assessed by the ‘Motors of influenza vaccination acceptance’ (MoVac-flu) and the ‘Motors of engagement with vaccination advocacy’ (MovAd) scales, respectively [36]. These two scales consist of 9 and 11 items and are measured on a 7-point Likert scale ranging from 1, “strongly disagree,” to 7, “strongly agree”. They assess the empowerment feelings of HCWs about the value (the flu jab plays a vital role in protecting my life and that of others), impact (vaccination greatly reduces my risk of catching the flu), knowledge (I understand how the flu jab helps my body fight the flu virus), and autonomy (I can choose whether to get a flu jab or not) about vaccinations (MoVac-flu) and the advocacy of vaccinations (MovAd). All the above indicative items in this paragraph come from the MoVac-flu scale. The Greek versions of the MoVac-flu and the MovAd scales have good psychometric properties [37,38]. Cronbach’s α internal consistency coefficients were α = 0.94 and α = 0.92, respectively, indicating excellent internal reliability.

#### 2.2.4. Other Predictors of Vaccination Behavior

Participants reported on smoking habits and the presence of chronic illnesses.

### 2.3. Statistical Analysis

Continuous variables are presented as means (standard deviation), while categorical variables are presented as percentages. The Kolmogorov–Smirnov test indicated the distribution of the continuous variables and the need for parametric methods. Bivariate analysis was performed between independent variables and vaccination status (chi-square test, chi-square trend test, and independent samples *t*-test). Subsequently, we used multivariate logistic regression models (odds ratios with 95% confidence intervals) to adjust for potential confounders. Independent variables significantly associated (*p* < 0.20) in bivariate analysis were entered into the backward stepwise multivariate logistic regression analysis with vaccination status as the dependent variable. Criteria for entry and removal of variables were based on the likelihood ratio test (enter/remove limits of *p* < 0.05 and *p* > 0.10, respectively). All two-sided statistical tests and *p*-values < 0.05 were considered statistically significant. Statistical analyses were performed using SPSS v.21.0 (IBM SPSS Statistics for Windows, Armonk, NY, USA: IBM Corp.).

### 2.4. Compliance with Ethical Standards

Ethical approval was obtained from all hospitals’ Ethical Review Boards. Participation in the study was voluntary and was performed according to the ethical standards of the Declaration of Helsinki as revised in Brazil. Informed consent was considered positive when participants returned the completed questionnaires.

## 3. Results

Of the 400 HCWs who received the survey, 296 responded. The mean age of HCWs was 43.3 years, and most participants were female (84.8%). Regarding vaccination status, 24.3% (*n* = 72) of the HCWs stated that they received the influenza vaccine this year, while 20.9% (*n* = 62) said they accept it yearly. Demographics and other characteristics are illustrated in Table 1.

Mean scores for subscales of the ‘Intention for vaccination uptake’ were above the mid-point (=3), indicating positive beliefs and an intention towards vaccination. In the same way, mean scores on the MoVac-flu scale, the MovAd scale, and the subscales were above the mid-point (=4), indicating high internal motivation and advocacy towards vaccination—Table 2.

Bivariate analysis between independent variables and vaccination status is shown in Table 3, while multivariate logistic regression analysis is shown in Table 4. According to the bivariate analysis, the percentage of vaccinated males was about twice as high as the percentage of females, while physicians were more frequently vaccinated than nurses and nurse assistants. Also, a higher age and educational level were related to a greater probability of vaccination. All scales of the Planned Behavior model (behavioral beliefs, normative beliefs, control beliefs, and intention score) were positively related to vaccination status. Vaccinated HCWs had higher scores on the MoVac-flu scale and the MovAd scale.

A multivariate analysis identified that increased intention score and age were associated with a greater probability of vaccination during this year (*p* < 0.001 and *p* = 0.016, respectively) and every year (*p* < 0.001 and *p* = 0.041, respectively). Also, an increased score on the MoVac-flu scale was associated with an increased probability of vaccination during this year (*p* < 0.001). HCWs with a chronic disease had a greater probability of vaccination during this year (*p* = 0.024) and every year (*p* = 0.002).

## 4. Discussion

The present study simultaneously tested the associations of two theories based on rational reasoning and motivation and empowerment with seasonal influenza vaccination in Greek HCWs. It showed that significant predictors of influenza vaccination uptake this year included an older age, the presence of chronic diseases, an increased score in intention (as conceptualized in the ‘TPB’), and an increased score in the motivation for vaccination (MoVac-flu scale, as conceptualized in the ‘CME’). In contrast, yearly influenza vaccination was associated with advanced age, chronic diseases, and intention scores. Thus, this an essential addition to the current literature for both theories. However, they have different starting points that focus on rational reason and emotions and provide valuable insights on influenza vaccination uptake in HCWs.

In the current study, only the MoVac-flu scale exhibited significant associations with vaccination in multivariable logistic models, which highlights the importance of considering cognitive empowerment as an important factor when implementing programs that aim to promote vaccination uptake in HCWs. In addition, this finding partially agrees with a recent study in six European countries in which increased engagement in influenza vaccine uptake was associated with higher scores on all MoVac-flu and MovAd subscales [35]. Still, there were some differences between the associations in the participating countries, implying the possible existence of other country-specific norms [39] that are related to vaccine uptake.

In the study of Dardalas et al. [37] in Greece of individuals over 60, no associations were reported between MoVac-flu and MovAd scales and influenza vaccination, and the intention score was its strongest predictor. It may be the case that healthcare professionals have different reasons for vaccination than the general population, but this notion should be further tested via empirical research. What is of interest in the study above is that the general population reported similar or higher scores in many MoVac-flu and MovAd subscales as the sample of HCWs in this study. More specifically, in this sample, the mean MoVac-flu scale total score, mean Value, and mean impact subscale scores were 4.7, 4.4, and 4.3, respectively, compared to 5.4, 5.5, and 4.8 in the sample of people over 60. There were no other prominent differences between the samples, except for the MovAd knowledge subscale, which was higher in HCWs. Older people who face more chronic diseases and have more frequent contact with the health care system are well-informed and empowered concerning SIV because influenza is likely a life-threatening condition for them. HCWs of younger age may have the professional responsibility to uptake and promote SIV. Still, they may not perceive it as threatening to themselves, which may explain the reported differences with the general population.

The overall low influenza vaccination uptake in this study is comparable to the national rate reported in a previous recent survey [40] and similar to the European rate (generally less than 30%) [7]. But it is lower than that reported for GPs in Greece [13] and the 40.3% reported for nurses in Ireland [25].

In bivariate analyses, demographic factors associated with HCWs’ intention to receive the influenza vaccination were male sex, older age, higher educational level, and being a physician. Other studies have also suggested that male HCWs have a higher likelihood of vaccine uptake [29]. In line with our results, similar studies [13,41,42] reported an increase in vaccine coverage with age. It may be that older individuals consider themselves more vulnerable to influenza complications, may have a positive experience of influenza vaccination from past years, and may feel a higher risk of severe health problems related to influenza, leading them to be vaccinated. Physicians get a vaccine shot more frequently than nurses, which aligns with many similar studies that could be explained mainly by nurses’ fear of vaccine safety [43]. However, the profession did not exhibit any significant association in the adjusted models. The Department of Work yielded no significant correlation with any study variables concerning influenza vaccination uptake. However, one would expect HCWs working in general internal medicine departments to be more “sensitive” in receiving the vaccine due to numerous interactions with patients and their relatives or HCWs working in ICUs to be alert since they treat patients with severe cases of influenza [44].

## 5. Limitations and Strengths

Several limitations must be considered when interpreting the results of this study. The cross-sectional research design does not permit any assumptions on causality. However, it is logical that past behaviors affect current ones and that the initial intentions are translated into behaviors and not vice versa. The questionnaires were self-administered, and thus recall or social desirability bias cannot be excluded. However, this is the primary approach to vaccine uptake research globally. The control tool beliefs subscale in the ‘TPB’ displayed a Cronbach α of 0.63, somewhat lower than the recommended minimum of 0.70, and this limitation should be noted as well. The study was conducted only in hospitals from a northern prefecture in Greece, and thus the generalizability of the results beyond this sample should be made with caution. Despite this, it is one of the first studies that tested vaccination uptake based on behavior using the ‘TPB’ and the ‘CME’.

## 6. Conclusions

Numerous factors, including increased age, intention score, motivation for vaccination (MoVac-flu) score, and the presence of chronic diseases, were associated with HCWs’ intention to get the seasonal influenza vaccination the last year and age, intention score, and the presence of chronic diseases for every year. Interventions focused on the variables identified in this study may be used to increase HCWs’ vaccination rates. This is an essential addition to the current literature because it pinpoints that vaccination uptake is simultaneously affected by logical cognitive processes and factors related to motivation and empowerment in distinct self-regulatory domains such as value, impact, knowledge, and autonomy. Future studies are required to assess the concept of vaccine literacy as an important determinant in the choice to be vaccinated, as it might pull HCWs out of vaccine hesitancy, thus increasing their engagement with vaccines.

## Figures and Tables

**Table 1 vaccines-11-01508-t001:** Demographic, professional, and behavioral characteristics of HCWs.

Characteristics	*n*	%
Gender		
Males	45	15.2
Females	251	84.8
Age	43.3 ^a^	8.7 ^b^
Marital status		
Singles/divorced/widows	83	28.0
Married	213	72.0
Children		
No	76	25.7
Yes	220	74.3
Occupation		
Physicians	37	12.5
Nurses	177	59.8
Nurse assistants	82	27.7
Department		
General internal medicine	201	71.5
ICUs	80	28.5
Educational level		
Secondary education	84	28.4
Tertiary education	179	60.5
MSc/PhD	33	11.1
Years of experience	16.2 ^a^	9.7 ^b^
Smoking		
No	196	66.2
Yes	100	33.8
Chronic disease		
No	269	90.9
Yes	27	9.1

^a^ Mean, ^b^ standard deviation.

**Table 2 vaccines-11-01508-t002:** Descriptive statistics for scales of TPB model, MoVac-flu scale, and MovAd scale.

Scale	Mean	Standard Deviation	Median	Minimum Value	Maximum Value	Cronbach’s Alpha
Behavioral beliefs	3.4	1.0	3.3	1	5	0.94
Normative beliefs	3.1	0.8	3.2	1	5	0.87
Control beliefs	3.8	0.5	3.8	2	5	0.63
Intention	3.0	1.1	3	1	5	0.95
MoVac-flu scale	4.7	1.3	4.6	1	7	0.92
Value	4.4	1.7	4.3	1	7	
Impact	4.3	1.7	4.3	1	7	
Knowledge	5.3	1.3	5.3	1	7	
Autonomy	6.1	1.4	7	1	7	
MovAd scale	4.8	1.2	4.9	1	7	0.92
Value	4.7	1.6	5	1	7	
Impact	4.6	1.5	4.7	1	7	
Knowledge	4.8	1.6	5	1	7	
Autonomy	5.4	1.4	6	1	7	

**Table 3 vaccines-11-01508-t003:** Bivariate analysis between independent variables and vaccination status.

	Vaccination This Year				*p*-Value	Vaccination Every Year				*p*-Value
	No		Yes			No		Yes		
	*n*	%	*n*	%		*n*	%	*n*	%	
**Gender**					**0.002 ^a^**					**0.009 ^a^**
Males	26	57.8	19	42.2		29	64.4	16	35.6	
Females	198	78.9	53	21.1		205	81.7	46	18.3	
Age	42.9 ^b^	8.7 ^c^	45.4 ^b^	8.4 ^c^	**0.03 ^d^**	42.8 ^b^	8.3 ^c^	45.4 ^b^	9.6 ^c^	**0.04 ^d^**
**Marital status**					0.9 ^a^					0.6 ^a^
Singles/divorced /widows	63	75.9	20	24.1		64	77.1	19	22.9	
Married	161	75.6	52	24.4		170	79.8	43	20.2	
**Children**					0.7 ^a^					0.9 ^a^
No	59	77.6	17	22.4		60	78.9	16	21.1	
Yes	165	75.0	55	25.0		174	79.1	46	20.9	
**Occupation**					**0.004 ^a^**					**0.007 ^a^**
Physicians	20	54.1	17	45.9		22	59.5	15	40.5	
Nurses	138	78.0	39	22.0		145	81.9	32	18.1	
Nurse assistants	66	80.5	16	19.5		67	81.7	15	18.3	
**Department**					0.6 ^a^					0.8 ^a^
ICUs	62	77.5	18	22.5		64	80.0	16	20.0	
General Internal medicine	150	74.6	51	25.4		158	78.6	43	21.4	
**Educational level**					**0.02 ^e^**					**0.006 ^e^**
Secondary education	68	81.0	16	19.0		69	82.1	15	17.9	
Tertiary education	137	76.5	42	23.5		148	82.7	31	17.3	
MSc/PhD	19	57.6	14	42.4		17	51.5	16	48.5	
**Years of experience**	16.5 ^b^	9.9 ^c^	17.5 ^b^	9.4 ^c^	0.4 ^d^	16.5 ^b^	9.9 ^c^	17.7 ^b^	9.4 ^c^	0.4 ^d^
**Smoking**					0.7 ^a^					0.9 ^a^
No	147	75.0	49	25.0		155	79.1	41	20.9	
Yes	77	77.0	23	23.0		79	79.0	21	21.0	
**Chronic disease**					**0.01 ^a^**					**0.002 ^a^**
No	209	77.7	60	22.3		219	81.4	50	18.6	
Yes	15	55.6	12	44.4		15	55.6	12	44.4	
**Behavioral beliefs**	3.2 ^b^	0.9 ^c^	4.2 ^b^	0.9 ^c^	**<0.001 ^d^**	3.2 ^b^	0.9 ^c^	4.4 ^b^	0.7 ^c^	**<0.001 ^d^**
**Normative beliefs**	3.0 ^b^	0.7 ^c^	3.6 ^b^	0.7 ^c^	**<0.001 ^d^**	3.0 ^b^	0.7 ^c^	3.7 ^b^	0.6 ^c^	**<0.001 ^d^**
**Control beliefs**	3.8 ^b^	0.5 ^c^	4.0 ^b^	0.5 ^c^	**0.04 ^d^**	3.8 ^b^	0.5 ^c^	3.9 ^b^	0.5 ^c^	**0.3 ^d^**
**Intention score**	2.6 ^b^	0.9 ^c^	4.1 ^b^	0.9 ^c^	**<0.001 ^d^**	2.7 ^b^	0.9 ^c^	4.2 ^b^	0.7 ^c^	**<0.001 ^d^**
**MoVac-flu scale**	4.3 ^b^	1.2 ^c^	5.9 ^b^	1.1 ^c^	**<0.001 ^d^**	4.4 ^b^	1.2 ^c^	6.0 ^b^	0.8 ^c^	**<0.001 ^d^**
**MovAd scale**	4.7 ^b^	1.2 ^c^	5.3 ^b^	1.1 ^c^	**<0.001 ^d^**	4.6 ^b^	1.2 ^c^	5.7 ^b^	0.9 ^c^	**<0.001 ^d^**

^a^ Chi-square test, ^b^ mean, ^c^ standard deviation, ^d^ independent samples *t*-test, and ^e^ chi-square trend test.

**Table 4 vaccines-11-01508-t004:** Multivariate logistic regression analysis with vaccination status as the dependent variable (no vaccination: reference category).

Dependent Variable *Independent variable*	Odds Ratio	95% Confidence Interval for Odds Ratio	*p*-Value
**Vaccination this year**			
Males vs. females	1.70	0.62 to 4.62	0.300
Age	1.05	1.01 to 1.10	0.016
Married vs. singles/divorced/widows	2.01	0.58 to 7.06	0.274
No children vs. children	1.29	0.33 to 5.02	0.712
Physicians vs. nursing staff	1.47	0.46 to 4.74	0.517
Educational level			
Tertiary education vs. secondary education	1.10	0.43 to 2.81	0.846
MSc/PhD vs. secondary education	1.36	0.34 to 5.49	0.664
Years of experience	1.04	0.97 to 1.10	0.280
Smokers vs. non smokers	1.25	0.54 to 2.88	0.605
Chronic disease vs. no	3.78	1.20 to 11.93	0.024
Behavioral beliefs	1.56	0.79 to 3.09	0.198
Normative beliefs	0.60	0.27 to 1.33	0.207
Control beliefs	1.70	0.73 to 3.98	0.220
Intention score	4.05	2.39 to 6.89	<0.001
Life Orientation Test	0.95	0.84 to 1.08	0.948
MoVac-flu scale	1.56	1.03 to 2.37	0.038
MovAd scale	0.86	0.54 to 1.38	0.532
**Vaccination every year**			
Males vs. females	1.68	0.53 to 5.29	0.377
Age	1.05	1.00 to 1.09	0.041
Married vs. singles/divorced/widows	1.21	0.29 to 5.00	0.796
No children vs. children	1.23	0.26 to 5.77	0.791
Physicians vs. nursing staff	0.59	0.15 to 2.35	0.461
Educational level			
Tertiary education vs. secondary education	0.73	0.26 to 2.09	0.559
MSc/PhD vs. secondary education	1.97	0.42 to 9.20	0.390
Years of experience	1.02	0.95 to 1.10	0.578
Smokers vs. non smokers	1.67	0.64 to 4.36	0.300
Chronic disease vs. no	6.79	2.04 to 22.61	0.002
Behavioral beliefs	2.07	0.93 to 4.58	0.074
Normative beliefs	0.78	0.33 to 1.87	0.578
Control beliefs	0.49	0.19 to 1.33	0.162
Intention score	7.27	4.36 to 12.13	<0.001
Life Orientation Test	1.11	0.96 to 1.29	0.151
MoVac-flu scale	1.56	0.84 to 2.89	0.160
MovAd scale	1.10	0.62 to 1.96	0.745

## Data Availability

The data that support the findings of this study are available on request from the corresponding author.

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
