# Peer review of "Intention, Motivation, and Empowerment: Factors Associated with Seasonal Influenza Vaccination among Healthcare Workers (HCWs)"

_vaccines, 2023, doi:10.3390/vaccines11091508_

Round 1

Reviewer 1 Report

The study examined intention, motivation, and empowerment towards seasonal influenza vaccination among healthcare workers. Overall, it is clearly written. However, some content need improvement.

Major comments

1.      In introduction, the author stated too much about health theories from the 6th to the 9th paragraph. Please shorten these paragraphs to be briefer.

2.      In the 2nd paragraph of Discussion, MoVac-flu scale exhibited significant associations with vaccination. Please add practical implications of the results to promote vaccination.

Minor comments

1.      In the 1st paragraph of Introduction, please change 500.000 to 500,000.

2.      In the 5th paragraph of Discussion, please cite this paper Associations between COVID-19 Vaccination and Behavioural Intention to Receive Seasonal Influenza Vaccination among Chinese Older Adults: A Population-Based Random Telephone Survey. Vaccines (Basel). 2023 Jul 6;11(7):1213. doi: 10.3390/vaccines11071213.  for the sentence “In line with our results, similar studies[11,41] reported an
increase in vaccine coverage with age.”

It is clear.

Author Response

The study examined intention, motivation, and empowerment towards seasonal influenza vaccination among healthcare workers. Overall, it is clearly written. However, some content need improvement.

 Major comments

  1. In introduction, the author stated too much about health theories from the 6thto the 9th Please shorten these paragraphs to be briefer.

Thank you for your comment. Necessary changes have been made to make paragraphs briefer.

  1. In the 2ndparagraph of Discussion, MoVac-flu scale exhibited significant associations with vaccination. Please add practical implications of the results to promote vaccination.

This was a very useful comment. We added the information that you suggested

Minor comments

  1. In the 1stparagraph of Introduction, please change 500.000 to 500,000.

Done

  1. In the 5thparagraph of Discussion, please cite this paper Associations between COVID-19 Vaccination and Behavioural Intention to Receive Seasonal Influenza Vaccination among Chinese Older Adults: A Population-Based Random Telephone Survey. Vaccines (Basel). 2023 Jul 6;11(7):1213. doi: 10.3390/vaccines11071213.  for the sentence “In line with our results, similar studies[11,41] reported an
    increase in vaccine coverage with age.”
  2.  

Done

Reviewer 2 Report

This is an well written and well structured manuscript. I do not have any major cocerns. The manuscript could be improved by shortenning the introduction. The author may shorten this section by focusing on the literature relevant to the objective of this manuscript.

The ethical aspects could go at the end of the method section after the analysis.

Author Response

This is an well written and well structured manuscript. I do not have any major cocerns. The manuscript could be improved by shortenning the introduction. The author may shorten this section by focusing on the literature relevant to the objective of this manuscript.

The ethical aspects could go at the end of the method section after the analysis.

Thank you very much for your suggestions. The ethical aspects section went after the analysis as suggested

Reviewer 3 Report

This is a well-written manuscript by Bellali and colleagues aimed at exploring intention, motivation, and empowerment toward vaccination and vaccination advocacy as contributing factors to seasonal influenza vaccination in healthcare workers (HCWs).

Their work is well presented, and I have no major concerns. However, I have some minor comments:

The authors state that vaccination is considered the most effective tool for managing the spread of influenza infection. It would be beneficial to include influential references to support this claim.

In the introduction, the authors refer to the previously unknown virus as COVID-19. It should be corrected to SARS-CoV-2, as COVID-19 is the name of the disease.

In the introduction, the authors mention that the focus on COVID-19 led to a lower demand for the influenza vaccine and a significant decline in flu vaccination. However, it's important to note that this statement is not entirely accurate, as some important reports (e.g. [1]) showed an increase in coverage rates in certain groups during the first winter season after COVID-19. Please address this discrepancy.

The introduction is somewhat wordy; I suggest reducing its length.

The methods and results sections are sound and well presented.

While the authors may not have assessed the level of vaccine literacy among healthcare workers, it has been suggested that vaccine literacy is an important determinant in the choice to get vaccinated. I recommend briefly introducing this concept in the discussion, providing a proper definition, and interpreting the specific results of this study in light of it.

1. https://pubmed.ncbi.nlm.nih.gov/34915972/

Minor editing of English language required

Author Response

This is a well-written manuscript by Bellali and colleagues aimed at exploring intention, motivation, and empowerment toward vaccination and vaccination advocacy as contributing factors to seasonal influenza vaccination in healthcare workers (HCWs).

Their work is well presented, and I have no major concerns. However, I have some minor comments:

The authors state that vaccination is considered the most effective tool for managing the spread of influenza infection. It would be beneficial to include influential references to support this claim.

We added references to support the claim

In the introduction, the authors refer to the previously unknown virus as COVID-19. It should be corrected to SARS-CoV-2, as COVID-19 is the name of the disease.

Thank you for the comment. We corrected in text as suggested

In the introduction, the authors mention that the focus on COVID-19 led to a lower demand for the influenza vaccine and a significant decline in flu vaccination. However, it's important to note that this statement is not entirely accurate, as some important reports (e.g. [1]) showed an increase in coverage rates in certain groups during the first winter season after COVID-19. Please address this discrepancy.

Thank you for the comment. We addressed the discrepancy.

The introduction is somewhat wordy; I suggest reducing its length.

We reduced the length of the introduction

The methods and results sections are sound and well presented.

While the authors may not have assessed the level of vaccine literacy among healthcare workers, it has been suggested that vaccine literacy is an important determinant in the choice to get vaccinated. I recommend briefly introducing this concept in the discussion, providing a proper definition, and interpreting the specific results of this study in light of it.

https://pubmed.ncbi.nlm.nih.gov/34915972/

Thank you very much for the useful comment. We added the importance of vaccine literacy in the conclusion section as a concept for future studies.